# REPRESENTATION DISENTANGLEMENT IN GENERATIVE MODELS WITH CONTRASTIVE LEARNING

## ABSTRACT

Contrastive learning has shown its effectiveness in image classification and generation. Recent works apply contrastive learning on the discriminator of the Generative Adversarial Networks. However, there is little work exploring if contrastive learning can be applied to the encoders-decoder structure to learn disentangled representations. In this work, we propose a simple yet effective method via incorporating contrastive learning into latent optimization, where we name it **ContraLORD**. Specifically, we first use a generator to learn discriminative and disentangled embeddings via latent optimization. Then an encoder and two momentum encoders are applied to dynamically learn disentangled information across a large number of samples with content-level and residual-level contrastive loss. In the meanwhile, we tune the encoder with the learned embeddings in an amortized manner. We evaluate our approach on ten benchmarks in terms of representation disentanglement and linear classification. Extensive experiments demonstrate the effectiveness of our ContraLORD on learning both discriminative and generative representations.

## 1 INTRODUCTION

In recent years, disentanglement of factors in images has attracted many researchers' attention, which mainly includes two folds: adversarial and non-adversarial methods. Adversarial methods (Mathieu et al., 2016; Denton & Birodkar, 2017; Hadad et al., 2018; Razavi et al., 2019; Gabbay & Hoshen, 2021) often apply a min-max optimization framework (Goodfellow et al., 2014) for disentanglement of images, which costs much time on hyper-parameters tuning. In terms of non-adversarial models, several variational autoencoders (Higgins et al., 2017; Kim & Mnih, 2018) variants have been proposed to disentangle the generative factors in an unsupervised manner without inductive biases, which did not achieve satisfactory results as proven in an empirical study (Locatello et al., 2019).

With the extra class supervision, semi-supervised methods achieve promising performance in disentanglement. Typically, comprehensive experiments in (Locatello et al., 2020) validate the effectiveness of a limited amount of supervision in state-of-the-art unsupervised disentanglement models. LORD (Gabbay & Hoshen, 2020) applies a latent optimization framework with a noise regularizer on content embeddings to achieve superior performance over amortized inference. Based on LORD, OverLORD (Gabbay & Hoshen, 2021) is proposed to disentangle class, correlated and uncorrelated attributes for image translation. A more recent work (Gabbay et al., 2021) adopts a pre-trained CLIP (Radford et al., 2021) model to generate partial annotations for image manipulation. However, there exist two main drawbacks among these methods: 1) using different separate encoders for different factors is resource-wasteful for real-world applications and requires expensive human design. 2) just learning the content embeddings inside each sample is not sufficient to learn the diversity existing in the dataset.

Driven by the shortcomings discussed above, we propose a simple yet effective method named ContraLORD, where we incorporate contrastive learning into latent optimization for representation disentanglement. Recent works (Deng et al., 2020; Ojha et al., 2020) apply the contrastive learning on the discriminator of the GAN (Goodfellow et al., 2014) for disentangling representations. Typically, the 3D imitative-contrastive learning in (Deng et al., 2020) is used for controllable face image generation by comparing pairs of generated images. However, in this work, we focus on applying

contrastive learning on the encoders to learn the discriminative and generative embeddings with disentangled information. Specifically, we first apply a generator to learn discriminative and generative embeddings via latent optimization. Then we apply an encoder and a momentum encoder to dynamically learn disentangled information across a large number of samples with content-level and residual-level contrastive loss. In the meanwhile, we use the learned discriminative and generative embeddings to tune the encoder in an amortized manner.

We evaluate our ContraLORD on two main tasks: linear classification and disentanglement. Extensive experiments show the effectiveness of the learned discriminative embeddings on linear classification and generative embeddings on the disentanglement of factors. We conduct comprehensive studies on three benchmarks on linear classification and seven benchmarks on disentanglement to investigate if contrastive self-supervised models can learn disentangled features. In the meantime, we achieve superior performance on linear classification compared to baselines. Our ContraLORD also achieves promising results over state-of-the-art methods in terms of disentanglement.

The main contributions of this work can be summarized as follows:

- We present a simple yet effective method called ContraLORD by incorporating contrastive learning into latent optimization for representation disentanglement and linear classification.
- We formally explore the disentangled features across a large number of samples with content-level and residual-level contrastive losses.
- Extensive experiments on ten benchmarks show the effectiveness of our approach on learning disentangled representations.

## 2 RELATED WORK

**Discriminative representations learning.** Discriminative representations learning has addressed researchers' attention for a long time since discriminative representations are significant for image classification. Most of the previous works adopt supervised (Lezama et al., 2018) and unsupervised learning (Ji et al., 2017; Zhang et al., 2018; Zhou et al., 2018; Zhang et al., 2019) to learn embeddings that most discriminate between classes in the dataset. Typically, the principle of maximal coding rate reduction (Yu et al., 2020) is applied to maximize the coding rate difference between the whole dataset and the sum of each separate class. However, there exists little work of contrastive learning to explore the discriminative representations for the pre-training stage. In this work, we mainly focus on learning discriminative embeddings for linear classification by incorporating contrastive learning into latent optimization to improve the performance of baselines.

**Disentangled Representations Learning.** Disentangled representation learning aims at learning generative factors existing in the dataset, that is, disentanglement learning. A bunch of previous work focuses on unsupervised learning with variational autoencoders, such as $\beta$-VAE (Higgins et al., 2017), Factor-VAE (Kim & Mnih, 2018). Following those work, DCI disentanglement (Eastwood & Williams, 2018), SAP score (Kumar et al., 2018), and Mutual Information Gap (MIG) (Chen et al., 2018) are often utilized as quantitative metrics to measure the quality of disentangled representations. In recent years, semi-supervised models have been used by many researchers in the literature. Adding a limited amount of supervision to unsupervised models is proven in (Locatello et al., 2020) to be effective in learning disentangled representations for real-world scenarios. LORD (Gabbay & Hoshen, 2020) leverages the latent optimization framework with a noise regularizer on content embeddings for class and content disentanglement. More recently, a simple framework for disentangling labeled and unlabeled attributes is utilized in OverLORD (Gabbay & Hoshen, 2021) for high-fidelity images synthesis. A study (Gabbay et al., 2021) uses a CLIP (Radford et al., 2021) pre-trained model to annotate a set of attributes for disentangled image manipulation. In this work, we intend to learn disentangled embeddings via combining latent optimization and contrastive self-supervised learning.

**Contrastive Learning.** Recently, contrastive self-supervised learning (Tian et al., 2020; Chen et al., 2020a;b; Grill et al., 2020; He et al., 2020; Chen et al., 2020c; Chen & He, 2021) has been explored a lot by many effective methods. SimCLR (Chen et al., 2020a), an end-to-end structure, is proposed to pull away from the features of each instance from those of all other instances in the training set. In the self-supervised setting, low-level image transformations such as cropping, scaling, and color

jittering are utilized for encoding the in-variances from samples. The InfoNCE loss, that is, the normalized temperature-scaled cross-entropy loss, are often optimized to maximize the similarity between positive samples and minimize the similarity between negative samples. Large batch size is always used in this kind of end-to-end structure (Chen et al., 2020a;b) to accumulate a large bunch of negative samples in the contrastive loss. PIRL (Misra & Maaten, 2020) without a large batch size applies a memory bank to store negative samples and update representations at a specified stage. MoCo (He et al., 2020) and MoCov2 (Chen et al., 2020c) replace the memory bank with a memory encoder to queue new batch samples and to dequeue the oldest batch. In this work, we leverage content-level and residual-level momentum encoders to store a queue of negative samples with disentangled information for learning generative embeddings, where content-level and residual-level contrastive losses are applied to capture content and residual representations.

## 3 METHODOLOGY

### 3.1 PROBLEM SETUP

In this part, we first begin with the problem setup, and formally define the notations for easy reading. In terms of the problem, our goal is to demystify the disentangled and discriminative features learned by contrastive self-supervised learning. To address this problem, we propose a simple yet effective method by combining contrastive learning and latent optimization together for representation disentanglement. To explain it better, we define the notations below in a unified manner.

**Notations.** Given a set of training examples $\mathcal{X} = \{\mathbf{x}_1, \mathbf{x}_2, \cdots, \mathbf{x}_n\}$. For each image $\mathbf{x}_i, i \in \{1, 2, \cdots, n\}$, we need to learn one discriminative embeddings $\mathbf{d}$ and a generative embeddings $\mathbf{g}_i$ from a pre-defined set of embeddings $\{\mathbf{g}_1, \mathbf{g}_2, \cdots, \mathbf{g}_m\}$, where $m$ denotes the total number of generative factors in the training data. That is, $\mathbf{d}_i \in \mathbb{R}^{1 \times d}, \mathbf{g}_i \in \mathbb{R}^{1 \times g}$, where $d, g$ denote the dimensionality of discriminative and generative embedding, separately. In our setting, we split the generative embedding $\mathbf{g}_i$ into two folds: content embeddings $\mathbf{g}_i^c$ and residual embeddings $\mathbf{g}_i^r$. The content embeddings contain the information that is related to the discriminative embedding, while the residual embedding includes the uncorrelated information.

Overall, the objective of our work is to learn $\mathbf{d}_i, \mathbf{g}_i^c, \mathbf{g}_i^r$ for each image $\mathbf{x}_i$ from a training dataset. In the next part, we present the technical details of our method. To learn $\mathbf{d}_i, \mathbf{g}_i^c, \mathbf{g}_i^r$ from a set of training examples, we propose a simple yet effective approach called ContraLORD, as shown in Figure 1. Our ContraLORD mainly includes two parts: 1) embedding optimization: we first use a generator $G(\cdot)$ to learn discriminative and disentangled embeddings via latent optimization. 2) encoder pre-training: we apply a encoder $f(\cdot)$ and a momentum encoder $f_m(\cdot)$ to dynamically learn disentangled information across large amount of samples with content- and residual-level contrastive loss.

### 3.2 EMBEDDING OPTIMIZATION

In order to learn the discriminative and disentangled embeddings, we are motivated by LORD (Gabbay & Hoshen, 2020) to introduce the latent optimization in the first stage. Specifically, we apply a generator $G(\cdot)$ to reconstruct the original image $\mathbf{x}_i$ by using the discriminative and disentangled embeddings of each sample. Instead of using the KL-divergence in variation auto-encoders (Kingma & Welling, 2014), we equally add a regularizer with Gaussian noise of a fixed variance to the disentangled embeddings $\mathbf{g}_i^c$ and $\mathbf{g}_i^r$. Thus, the objective of embeddings optimization is defined as

$$\mathcal{L}_{opt} = s \sum_{i=1}^{n} \ell(G(\mathbf{d}_i, \mathbf{g}_i^c + \mathbf{z}_i, \mathbf{g}_i^r + \mathbf{z}_i)), \mathbf{x}_i) + \lambda \cdot (||\mathbf{g}_i^c||^2 + ||\mathbf{g}_i^r||^2) \tag{1}$$

where $\ell(\cdot)$ denotes $\ell_2$ loss for synthetic data and VGG perceptual loss for real images. $\lambda$ is the penalty weight of the capacity of the disentangled embeddings. $\mathbf{z}_i \sim \mathcal{N}(\mathbf{0}, \sigma^2 I)$. In this way, we can learn the disentangled embeddings $\mathbf{d}_i, \mathbf{g}_i^c, \mathbf{g}_i^r$ without any adversarial learning involved, that is, $\widetilde{\mathbf{d}}_i, \widetilde{\mathbf{g}}_i^c, \widetilde{\mathbf{g}}_i^r = \arg\min \mathcal{L}_{opt}$. For training sets with annotations, $\widetilde{\mathbf{d}}_i$ is given.

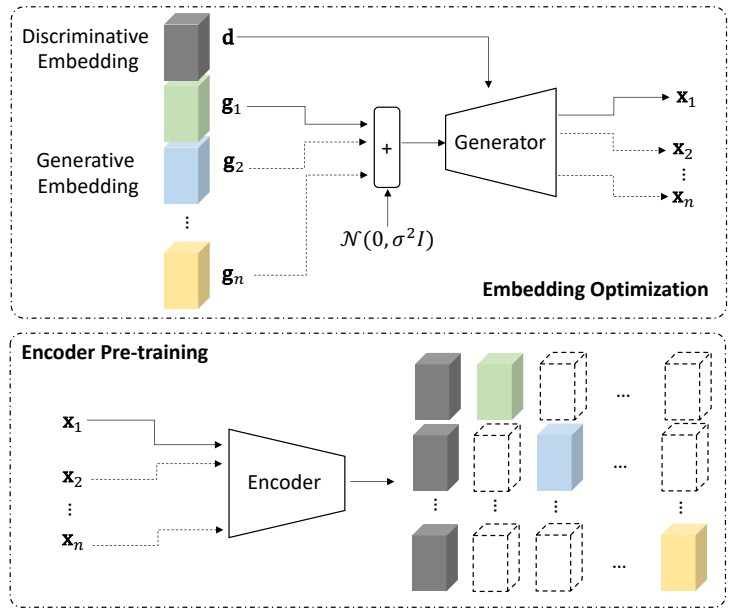

Figure 1: The overall framework of our proposed ContraLORD model.

### 3.3 ENCODER PRE-TRAINING

After learning the optimized embeddings, we need to train a generalized encoder during the pre-training stage. In order to optimize the encoder $f(\cdot)$, we reconstruct the original image $\mathbf{x}_i$ with the output embeddings from $f(\cdot)$, and the loss function is calculated as

$$\mathcal{L}_{rec} = \sum_{i=1}^{n} \ell\bigg( G\Big(h_{\mathbf{d}}\big(f(\mathbf{x}_i)\big), h_{\mathbf{g}}^c\big(f(\mathbf{x}_i)\big), h_{\mathbf{g}}^r\big(f(\mathbf{x}_i)\big)\Big), \mathbf{x}_i\bigg) \tag{2}$$

where $\ell(\cdot)$ denotes $\ell_2$ loss for synthetic data and VGG perceptual loss for real images. $h_{\mathbf{d}}(\cdot), h_{\mathbf{g}}^c(\cdot), h_{\mathbf{g}}^r(\cdot)$ denote the head for generating the discriminative, content-level, and residual-level embeddings. To learn more disentangled information in the discriminative and disentangled embeddings, we use the learned embeddings in the first stage as the supervision and define the objective as:

$$\mathcal{L}_{sup} = \sum_{i=1}^{n} ||h_{\mathbf{d}}(f(\mathbf{x}_i)) - \widetilde{\mathbf{d}}_i||^2 + ||h_{\mathbf{g}}^c(f(\mathbf{x}_i)) - \widetilde{\mathbf{g}}_i^c||^2 + ||h_{\mathbf{g}}^r(f(\mathbf{x}_i)) - \widetilde{\mathbf{g}}_i^r||^2 \tag{3}$$

where $\widetilde{\mathbf{d}}_i, \widetilde{\mathbf{g}}_i^c, \widetilde{\mathbf{g}}_i^r$ denote the learned representations from the first optimization stage, respectively.

**Content-level Contrastive Loss.** To learn more disentangled content embeddings across examples in the dataset, we feed a content momentum queue $\{\mathbf{x}_1^{key}, \mathbf{x}_2^{key}, \cdots, \mathbf{x}_k^{key}, \cdots, \mathbf{x}_K^{key}\}$ of one query sample $\mathbf{x}_i^{query}$ into a momentum encoder $f_m(\cdot)$. The illustration of the content-level contrastive learning is shown in Figure 2 (left). As can be seen, we generate the content embeddings $\mathbf{g}_k^c, k \in \{1, 2, \cdots, K\}$ from the momentum queue, and $\mathbf{g}_q^c$ from the query sample $\mathbf{x}_i^{query}$. Then we calculate the similarity between the original content embedding $\mathbf{g}_i^c$ and $\mathbf{g}_q^c, \mathbf{g}_k^c$ for content-level contrastive loss. Finally, the content-level contrastive loss is defined as

$$\mathcal{L}_{con} = \sum_{i=1}^{n} -\log \frac{\exp(\mathbf{g}_i^c \cdot \mathbf{g}_q^c / \tau)}{\exp(\mathbf{g}_i^c \cdot \mathbf{g}_q^c / \tau) + \sum_{k=1}^{K} \exp(\mathbf{g}_i^c \cdot \mathbf{g}_k^c / \tau)} \tag{4}$$

where $K$ denotes the number of negative samples in the momentum queue. $\tau$ is a temperature hyper-parameter. In the backward process, we update the parameters of the encoder $f(\cdot)$ according

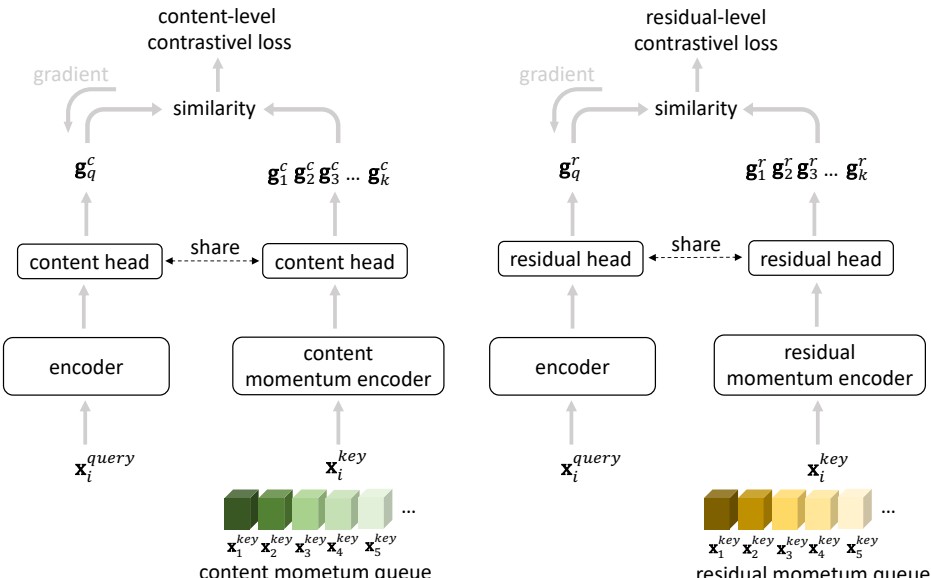

Figure 2: Content-level (**left**) and residual-level (**right**) contrastive learning designed in our ContraLORD.

to the gradient of this loss. The parameters of the momentum encoder $f_m^c(\cdot)$ is updated by $f_m(\cdot) \leftarrow m f_m(\cdot) + (1-m)f(\cdot)$, where $m \in (0,1]$ is a momentum coefficient.

**Residual-level Contrastive Loss.** In order to further disentangle the residual part in the disentangled embeddings, we introduce a residual momentum encoder $f_m^r(\cdot)$ to receive the residual momentum queue and generate a set of residual embeddings $\mathbf{g}_k^r, k \in \{1, 2, \cdots, K\}$. Thus, the residual-level contrastive loss is defined with key embeddings $\mathbf{g}_k^r$, query embedding $\mathbf{g}_q^r$, and the original embedding $\mathbf{g}_i^r$ as

$$\mathcal{L}_{res} = \sum_{i=1}^{n} -\log \frac{\exp(\mathbf{g}_i^r \cdot \mathbf{g}_q^r / \tau)}{\exp(\mathbf{g}_i^r \cdot \mathbf{g}_q^r / \tau) + \sum_{k=1}^{K} \exp(\mathbf{g}_i^r \cdot \mathbf{g}_k^r / \tau)} \tag{5}$$

where $K$ denotes the number of negative samples in the momentum queue, and $\tau$ is a temperature hyper-parameter. The gradient of this loss is also used to update the parameters of the encoder $f(\cdot)$. We update the parameters of the momentum encoder $f_m(\cdot)$ by $f_m(\cdot) \leftarrow m f_m(\cdot) + (1-m)f(\cdot)$, where $m \in (0,1]$ is a momentum coefficient.

The overall objective of our ContraLORD is optimized in an end-to-end manner as

$$\mathcal{L} = (\mathcal{L}_{rec} + \cdot\mathcal{L}_{sup}) + \lambda_{con} \cdot \mathcal{L}_{con} + \lambda_{res} \cdot \mathcal{L}_{res} \tag{6}$$

where $\lambda_{con}, \lambda_{res}$ denote the weight of the content-level and residual-level contrastive loss, respectively. Extensive ablation studies are conducted to explore the effects of each loss on the final performance of our ContraLORD. We summarize the overall algorithm of our training approach in Algorithm 1.

### 3.4 SMOOTHNESS OF EMBEDDINGS

In order to measure the smoothness of embeddings pre-trained by the encoder, we borrow the idea of uniformity property in the instance-wise contrastive learning, and introduce the Gaussian potential kernel (Bartók & Csányi, 2015) to calculate the average pairwise Gaussian potential as:

$$smoothness = \mathbb{E}_{(\mathbf{g}_i, \mathbf{g}_j) \sim p_c}[e^{-t||\mathbf{g}_i - \mathbf{g}_j||^2}] + \mathbb{E}_{(\mathbf{g}_i, \mathbf{g}_j) \sim p_r}[e^{-t||\mathbf{g}_i - \mathbf{g}_j||^2}] \tag{7}$$

where $p_c, p_r$ denotes the distribution of content and residual embeddings in the hyper-sphere, and $t$ is a positive factor to define the weight of the $\ell_2$ distance between embeddings $\mathbf{g}_i$ and $\mathbf{g}_j$. In our experiments, we follow previous work (Wang & Isola, 2020) and set $t=2$.

---

**Algorithm 1** ContraLORD main learning algorithm

---

    **Input:** generator $G(\cdot)$, encoder $f(\cdot)$, momentum encoders $f_m^c(\cdot), f_m^r(\cdot)$, heads $h_{\mathbf{d}}(\cdot), h_{\mathbf{g}}(\cdot)$.
1: Initialize the parameters $G(\cdot), f(\cdot), f_m^c(\cdot), f_m^r(\cdot), h_{\mathbf{d}}(\cdot), h_{\mathbf{g}}(\cdot)$,
2: Initialize the embeddings $\mathbf{d}_i, \mathbf{g}_i, i \in \{1, 2, \cdots, n\}$
3:    # Embedding Optimization
4: **for** each epoch **do**
5:     Apply $G(\cdot)$ to reconstruct original images
6:     Calculate the optimization loss in Eq. 1
7:     Update $\mathbf{d}_i, \mathbf{g}_i$
8:    # Encoder Pre-training
9: **for** each epoch **do**
10:     Apply $f(\cdot), h_{\mathbf{d}}(\cdot), h_{\mathbf{g}}(\cdot)$ to reconstruct original images and calculate the loss in Eq. 2
11:     Apply embeddings $\mathbf{d}_i^*, \mathbf{g}_i^*$ as supervision and calculate the loss in Eq. 3
12:     Apply $f(\cdot), f_m^c(\cdot)$ to encode content features $\mathbf{g}_q^c, \mathbf{g}_k^c$ and calculate loss as in Eq. 4
13:     Apply $f(\cdot), f_m^r(\cdot)$ to residual features $\mathbf{g}_q^r, \mathbf{g}_k^r$ and calculate loss as in Eq. 5
14:     Compute the total loss in Eq. 6
15:     Update the parameters of $f(\cdot), h_{\mathbf{d}}(\cdot), h_{\mathbf{g}}(\cdot)$
16:     Update the momentum parameters of $f_m^c(\cdot), f_m^r(\cdot)$
17:     Update the content and residual momentum queue
    **Output:** $f(\cdot), h_{\mathbf{d}}(\cdot), h_{\mathbf{g}}(\cdot)$

---

# 4 EXPERIMENTS

## 4.1 DATASETS & CONFIGURATIONS

Following previous methods (He et al., 2020; Chen et al., 2020a), we evaluate the linear classification of the encoder pre-trained by our ContraLORD on three widely-used benchmarks, including CIFAR-10, CIFAR100, ImageNet-100 (Deng et al., 2009; Tian et al., 2020). In terms of disentanglement (Gabbay & Hoshen, 2020; 2021), we evaluate the disentangled embeddings on four synthetic: Shapes3D (Kim & Mnih, 2018), Cars3D (Reed et al., 2015), dSprites (Higgins et al., 2017), SmallNorb (LeCun et al., 2004); And three real datasets: CelebA (Liu et al., 2015), AFHQ (Choi et al., 2020), CelebA-HQ (Karras et al., 2018).

Specifically, Shapes3D (Kim & Mnih, 2018) contains 4 shapes, 8 scales, 15 orientations, 10 floor colors, 10 wall colors, and 10 object colors. Cars3D (Reed et al., 2015) includes 183 car CAD models with 24 different azimuth directions and 4 elevations, where 163 models are for training and 20 for testing. dSprites (Higgins et al., 2017) contains 3 shapes, 6 scales, 40 orientations, 32 x positions, and 32 y positions. SmallNorb (LeCun et al., 2004) consists of 50 toys with 5 generic classes, 6 lighting conditions, 9 elevations, and 18 azimuths. CelebA (Liu et al., 2015) includes 10,177 celebrities, in total 202,599 images, where we use 9,177 images for training and 1,000 images for testing. AFHQ (Choi et al., 2020) is an animal face dataset with 15,000 high-quality images of three categories: cat, dog, and wildlife. CelebA-HQ (Karras et al., 2018) contains 30,000 high-quality images from CelebA with gender as the class, and masks are used for the hairstyle disentanglement.

For a fair comparison, we follow the same setting as previous work (Gabbay & Hoshen, 2020; 2021). During embedding optimization, we set $d$=256, $g$=128, $K$=12800, and $\lambda$=0.001. The generator is optimized by Adam (Kingma & Ba, 2014) optimizer with a learning rate of 0.0001. We train the encoder with a learning rate of 0.001. For the regularized Gaussian noise, we set $\sigma$=1. For encoder pre-training, we closely follow MoCo (He et al., 2020) and use the same data augmentation. For the encoder networks, we experiment with the commonly used encoder architecture, ResNet-50. We train at batch size 256 for 200 epochs.

## 4.2 EVALUATION METRICS

For evaluation metrics, we use top-1, top-5 accuracy for linear classification. In terms of evaluating the disentangled embeddings, we use three mainly-used metrics in the literature: DCI (Eastwood & Williams, 2018), SAP score (Kumar et al., 2018), and MIG (Chen et al., 2018). DCI measures

the disentanglement, completeness, and informativeness of the generative embeddings. SAP score refers to a separated attribute predictability score that captures one generative factor in only one disentangled dimension. MIG is the mutual information gap to calculate the difference between the top two latent factors with the highest mutual information. In the meanwhile, we follow previous works (Gabbay & Hoshen, 2020; 2021) and evaluate our ContraLORD on FID and LPIPS. FID measures how the disentangled embeddings are translated to the target domain, while LPIPS is used for calculating the quality of transferred content embeddings in terms of perceptual similarity.

## 4.3 EXPERIMENTAL RESULTS

In this part, we conduct extensive experiments to evaluate the discriminative and disentangled embeddings learned by our ContraLORD, which demonstrates the advantage of our approach against previous work (Gabbay & Hoshen, 2020; 2021) to learn discriminative and disentangled representations via content-level and residual-level contrastive loss.

**Evaluation of discriminative embeddings.** We evaluate the quality of the discriminative embeddings on linear classification. Specifically, we train the linear model on frozen features from various self-supervised methods and report the experimental results in Table 1. Our ContraLORD substantially outperforms baselines (Gabbay & Hoshen, 2020; 2021) in terms of top-1 and top-5 accuracy on all benchmarks. Particularly, we achieve the performance gain over LORD (Gabbay & Hoshen, 2020) by 8.88%, 9.35%, 8.91%. In the meanwhile, we surpass a concurrent work (Gabbay & Hoshen, 2021) using the class embeddings with a higher dimension. This demonstrates the superiority of our ContraLORD incorporating content-level and residual-level contrastive learning into latent optimization. Furthermore, our ContraLORD outperforms the pure contrastive self-supervised methods (He et al., 2020; Chen et al., 2020a), which also validates the effectiveness of latent optimization in learning more generalized and discriminative embeddings for linear classification.

Table 1: Comparison results of linear classification on CIFAR-10, CIFAR-100, ImageNet-100, and TinyImageNet-200 datasets.

| Dataset | Method | Arch. | Epochs | Top-1 | Top-5 |
|---|---|---|---|---|---|
| CIFAR-10 | MoCo | ResNet-50 | 200 | 93.30 | 99.85 |
| | SimCLR | ResNet-50 | 200 | 92.00 | 99.81 |
| | LORD | ResNet-50 | 200 | 85.13 | 96.22 |
| | OverLORD | ResNet-50 | 200 | 91.62 | 98.61 |
| | ContraLORD (ours) | ResNet-50 | 200 | **94.01** | **99.89** |
| CIFAR-100 | MoCo | ResNet-50 | 200 | 71.70 | 90.23 |
| | SimCLR | ResNet-50 | 200 | 71.58 | 90.11 |
| | LORD | ResNet-50 | 200 | 63.32 | 87.05 |
| | OverLORD | ResNet-50 | 200 | 69.96 | 89.53 |
| | ContraLORD (ours) | ResNet-50 | 200 | **72.67** | **90.97** |
| ImageNet-100 | CMC | ResNet-50 | 200 | 66.20 | 88.75 |
| | MoCo | ResNet-50 | 200 | 72.80 | 91.04 |
| | LORD | ResNet-50 | 200 | 67.32 | 89.26 |
| | OverLORD | ResNet-50 | 200 | 70.16 | 90.45 |
| | ContraLORD (ours) | ResNet-50 | 200 | **76.23** | **92.52** |

**Evaluation of disentangled embeddings.** Following existing work (Gabbay & Hoshen, 2020; 2021), we evaluate the disentanglement performance of disentangled embeddings with 100 labels on four synthetic datasets. Table 2 reports the comparison results. As can be seen, our ContraLORD still outperforms existing methods in terms of all metrics, including DCI, SAP, and MIG. This shows that our ContraLORD with the content-level and residual-level contrastive loss is superior to learn more disentangled information in the disentangled embeddings. In the meanwhile, we follow the setting in OverLORD (Gabbay & Hoshen, 2021) and conduct experiments on three real benchmarks in Table 3. We can observe that our ContraLORD achieves the best performance in terms of 5 out of 7 evaluation metrics. For the other two metrics, we still achieve comparable results when compared to OverLORD (Gabbay & Hoshen, 2021). These results further validate the effectiveness of our ContraLORD in learning disentangled representations with more disentangled information.

**Smoothness of content and residual embeddings.** We simultaneously measure the smoothness score of content, and residual embeddings pre-trained by the encoder and report the results in the

Table 2: Disentanglement performance on Shapes3D, Cars3D, dSprites, and SmallNorb datasets.

| Dataset | Method | D (↑) | C (↑) | I (↑) | SAP (↑) | MIG (↑) | Smoothness (↑) |
|---|---|---|---|---|---|---|---|
| Shapes3D | Locatello *et al.* | 0.03 | 0.03 | 0.22 | 0.01 | 0.02 | 0.15 |
| | LORD | 0.54 | 0.54 | 0.54 | 0.15 | 0.43 | 0.48 |
| | Gabbay *et al.* | **1.00** | **1.00** | **1.00** | 0.30 | 0.96 | 0.82 |
| | ContraLORD (ours) | **1.00** | **1.00** | **1.00** | **0.42** | **1.00** | **0.96** |
| Cars3D | Locatello *et al.* | 0.11 | 0.17 | 0.22 | 0.06 | 0.04 | 0.16 |
| | LORD | 0.26 | 0.48 | 0.36 | 0.13 | 0.20 | 0.27 |
| | Gabbay *et al.* | 0.40 | 0.41 | 0.56 | 0.15 | 0.35 | 0.33 |
| | ContraLORD (ours) | **0.51** | **0.56** | **0.71** | **0.25** | **0.41** | **0.45** |
| dSprites | Locatello *et al.* | 0.01 | 0.01 | 0.16 | 0.01 | 0.01 | 0.12 |
| | LORD | 0.16 | 0.17 | 0.43 | 0.03 | 0.06 | 0.18 |
| | Gabbay *et al.* | 0.75 | 0.75 | 0.68 | 0.13 | 0.48 | 0.52 |
| | ContraLORD (ours) | **0.85** | **0.84** | **0.79** | **0.24** | **0.62** | **0.67** |
| SmallNorb | Locatello *et al.* | 0.02 | 0.08 | 0.18 | 0.01 | 0.01 | 0.13 |
| | LORD | 0.01 | 0.03 | 0.30 | 0.01 | 0.02 | 0.17 |
| | Gabbay *et al.* | 0.27 | 0.39 | 0.45 | 0.14 | 0.27 | 0.29 |
| | ContraLORD (ours) | **0.36** | **0.51** | **0.56** | **0.26** | **0.42** | **0.48** |

Table 3: Disentanglement performance on CelebA, AFHQ, and CelebA-HQ datasets.

| Method | CelebA | | | AFHQ | | CelebA-HQ | |
| | Id (↑) | Exp (↓) | Pose (↓) | FID (↓) | LPIPS (↑) | FID (F2M,↓) | FID (M2F,↓) |
|---|---|---|---|---|---|---|---|
| LORD | 0.48 | 3.2 | 3.5 | 97.1 | 0 | - | - |
| OverLORD | **0.63** | 2.7 | 2.5 | 16.5 | 0.51 | **54.0** | 42.9 |
| ContraLORD (ours) | 0.61 | **2.6** | **2.3** | **15.8** | **0.53** | 54.2 | **42.6** |

last column of Table 2. We can observe that our ContraLORD outperforms existing methods by a large margin (0.14, 0.12, 0.15, 0.19) on all four benchmarks in terms of the smoothness score, which shows the advantage of our ContraLORD on learning disentangled embeddings that are more uniformly distributed on the hyper-sphere. In the meanwhile, our smoothness score is positively correlated with the previous disentanglement metrics. This demonstrates the effectiveness of learning uniformly distributed embeddings of disentangled information for representations disentanglement.

## 5 ABLATION STUDY

In this section, we perform comprehensive ablation studies to explore the effect of each loss ($\mathcal{L}_{rec}, \mathcal{L}_{sup}, \mathcal{L}_{con}, \mathcal{L}_{res}$), batch size, and the number of negative samples ($K$) on the final performance of our ContraLORD. Unless specified, we conduct all ablation studies on the Shapes3D dataset.

Table 4: Ablation study on each loss.

| $\mathcal{L}_{rec}$ | $\mathcal{L}_{sup}$ | $\mathcal{L}_{con}$ | $\mathcal{L}_{res}$ | D (↑) | C (↑) | I (↑) | SAP (↑) | MIG (↑) | Smoothness (↑) |
|---|---|---|---|---|---|---|---|---|---|
| ✗ | ✗ | ✗ | ✗ | 0.02 | 0.01 | 0.13 | 0.01 | 0.01 | 0.09 |
| ✓ | ✗ | ✗ | ✗ | 0.26 | 0.29 | 0.31 | 0.08 | 0.15 | 0.22 |
| ✓ | ✓ | ✗ | ✗ | 0.54 | 0.54 | 0.54 | 0.15 | 0.42 | 0.48 |
| ✓ | ✓ | ✓ | ✗ | 0.91 | 0.89 | 0.88 | 0.37 | 0.85 | 0.75 |
| ✓ | ✓ | ✓ | ✓ | **1.00** | **1.00** | **1.00** | **0.42** | **1.00** | **0.96** |

**Effect of each loss.** To explore how each proposed loss affects the final performance of our method, we ablate each module on the final loss and show the disentanglement results in Table 4. Without four losses in the encoder pre-training stage, we achieve the worst performance. Adding $\mathcal{L}_{sup}$ to only $\mathcal{L}_{rec}$ boosts the results by 0.24, 0.28, 0.18, 0.07, 0.14, and 0.13. By combining $\mathcal{L}_{con}$ with $\mathcal{L}_{sup}$ and $\mathcal{L}_{rec}$, we achieve a performance gain of 0.37, 0.35, 0.34, 0.22, 0.43, and 0.27. These results demonstrate the effectiveness of our content-level and residual-level loss in learning disentangled embeddings. Finally, our ContraLORD, with all losses, achieves the best performance in terms of

all disentanglement metrics and the smoothness score, which validates the rationality of each loss on learning disentangled representations.

**Effect of batch size.** Table 5 reports the exploration study results of batch size. Specifically, we vary the batch size from 16, 32, 64, 128, 256, 512 during the encoder pre-training stage. From the results, we can observe that our approach performs the best when the batch size is 512. With a smaller batch size of 256, our ContraLORD does not have a large performance decline (0.03, 0.01) in terms of SAP score and Smoothness score. When the batch size is set to 32, our method has an obvious performance decrease, which shows the importance of suitable batch size in our content-level and residual-level contrastive loss by introducing negative samples across the same batch. When we increased the batch size to 1024, the performance of our approach on all disentanglement metrics and the smoothness score is deteriorated by the confusion of too many negative samples in the same batch.

Table 5: Ablation study on the bath size.

| Batch Size | D ($\uparrow$) | C ($\uparrow$) | I ($\uparrow$) | SAP ($\uparrow$) | MIG ($\uparrow$) | Smoothness ($\uparrow$) |
|---|---|---|---|---|---|---|
| 32 | 0.82 | 0.81 | 0.79 | 0.36 | 0.82 | 0.72 |
| 64 | 0.89 | 0.87 | 0.86 | 0.38 | 0.88 | 0.79 |
| 128 | 0.93 | 0.91 | 0.91 | 0.39 | 0.91 | 0.85 |
| 256 | **1.00** | **1.00** | **1.00** | 0.42 | **1.00** | 0.96 |
| 512 | **1.00** | **1.00** | **1.00** | **0.45** | **1.00** | **0.97** |
| 1024 | 0.99 | 0.98 | 0.98 | 0.41 | 0.99 | 0.93 |

**Effect of negative samples.** In order to explore the effect of negative samples on the final performance of our ContraLORD, we vary the number of negative samples from 1600, 3200, 6400, 12800, 25600, 51200 for the content and residual momentum queue. We show the experimental results in Table 6. As can be seen, with the increase of the number of negative samples in the momentum queue, our ContraLORD achieves better performance in terms of all metrics. However, too many negative samples, *i.e.*, a large number of negative samples, degrades the performance of our approach since it is hard for the content-level and residual-level contrastive loss to discriminate hard negative samples in the momentum queue with many negative samples. This further demonstrates the importance of negative samples in learning generative embeddings with disentangled information.

Table 6: Ablation study on the number of negative samples.

| $K$ | D ($\uparrow$) | C ($\uparrow$) | I ($\uparrow$) | SAP ($\uparrow$) | MIG ($\uparrow$) | Smoothness ($\uparrow$) |
|---|---|---|---|---|---|---|
| 1600 | 0.62 | 0.63 | 0.61 | 0.26 | 0.61 | 0.52 |
| 3200 | 0.81 | 0.79 | 0.77 | 0.33 | 0.79 | 0.68 |
| 6400 | 0.88 | 0.86 | 0.87 | 0.37 | 0.87 | 0.77 |
| 12800 | **1.00** | **1.00** | **1.00** | **0.42** | **1.00** | **0.96** |
| 25600 | 0.96 | 0.95 | 0.95 | 0.41 | 0.95 | 0.89 |
| 51200 | 0.87 | 0.85 | 0.85 | 0.36 | 0.86 | 0.75 |

## 6  CONCLUSION

In this work, we propose the ContraLORD, a simple yet effective approach by incorporating contrastive learning into latent optimization for representation disentanglement. Specifically, we first use a generator to learn discriminative and disentangled embeddings via latent optimization. Then an encoder and two momentum encoders are applied to dynamically learn disentangled information across a large number of samples with content-level and residual-level contrastive losses. Finally, we tune the encoder with the learned embeddings in an amortized manner. We conduct extensive experiments on ten benchmarks to demonstrate the effectiveness of our ContraLORD on learning disentangled representations. Comprehensive ablation studies also validate the rationality of each contrastive loss proposed in our approach. We also empirically observe the importance of negative samples across a large number of samples in learning generative embeddings with disentangled information.

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
