# OpenReview forum: "Representation Disentanglement in Generative Models with Contrastive Learning"
_ICLR.cc/2022/Conference — ICLR 2022 Submitted_

### Official Review · Reviewer_Th2p · 2021-10-31

**Correctness:** 4
**Technical Novelty And Significance:** 2
**Empirical Novelty And Significance:** 3
**Recommendation:** 8
**Confidence:** 3

**Main Review:**

Pros:
- The paper is well-written and is easy to follow and it addresses an important problem related to representation learning for generative models which is a popular topic in the community.

- The results validate the efficacy of the proposed method on many datasets/metrics.

- The contrastive learning method is interesting and is well-motivated in the sense that it is not surprising that it should help improve the latent representation, and I applaud the authors for making it work for the latent representation learning of a generative model.

Cons:
- The algorithm has many moving components such as generators, encoders, momentum encoders, and heads interacting in ways that seem complicated. Complicated methods are more difficult to reproduce and to use for new problem domains or build on for future research.

- Further, no code is provided to understand the implementation or help in reproducing/verifying the results.

- The novelty is slightly limited since it is direct integration of contrastive learning to OverLORD, which already performs latent optimization with disentanglement.

- Getting 100% performance in Table 4 by just adding the residual-level loss seems suspicious, can the authors explain that this result is not an anomaly?

- No multiple runs are shown to display the standard deviation and the mean result across random initializations. This result is important as it allows us to see if the reported performance is significant and robust to different seeds.

- How long did the training take? and how did the authors perform early stopping?

- No robustness test showing how the weight coefficients affect the residual and content-level loss terms.

**Summary Of The Paper:**

The authors propose a new method called ContraLord for the task of representation disentanglement for generative models. Their method extends LORD, a latent optimization method for improving content embeddings, and OverLORD which performs disentanglement on embeddings, by incorporating contrastive learning. Specifically, their method includes content-level and residual-level contrastive loss that leverage negative examples in a batch.  Their experimental results show promising results on standard datasets like CIFAR and ImageNet in terms of classification performance and disentanglement. The authors also included interesting ablations to showcase the efficacy of each loss term like the residual-level contrastive loss.

**Summary Of The Review:**

I recommend an accept since the method seems sound, well-motivated and reasonable and the results seem promising in difficult datasets. However, the novelty of this work is slightly lacking since it is a straightforward integration of a contrastive learning method (which have been a common theme recently) to OverLORD, the cited paper that performs latent optimization with disentanglement.

---

> ### Author Response · Authors · 2021-11-23
> **Response**
>
> We appreciate the insightful feedback, below we address the major concerns mentioned in the comments.
>
> Q1: Novelty.
>
> The main contribution of this work is to incorporate contrastive learning into latent optimization for both representation disentanglement and linear classification. We focus on demonstrating the effectiveness of contrastive learning in the disentanglement scenario, but not designing a working system. Therefore, we choose the LORD as a strong baseline and employ the most commonly used contrastive learning framework. On the other hand, we also reform the framework to leverage content-level and residual-level contrastive losses to capture content and residual representations across a large number of samples.
>
> Q2: Confusion about the result in Table 4.
>
> This result in Table 4 is not an anomaly. The residual-level loss is designed to the residual embeddings except for the disentangled factors, which helps to improve the disentanglement, completeness, and informativeness of learned content embeddings.
>
> Q3: Multiple runs.
>
> Due to limited computation resources, we did not run multiple times with various random seeds.
>
> Q4: Training time.
>
> For training time, we have two stages. With 8 V100-32G GPUs, we have 52 hours on the first stage and 71 hours on the second stage on the ImageNet-100 benchmark with 120k images.
>
> Q5: Robustness of weight coefficients.
>
> We will add more ablation analysis on the effect of weight coefficients on the residual and content-level loss terms in future work. In our experiments, we set them to 1 in default.

---

> > ### Comment · Reviewer_Th2p · 2021-11-29
> > **Thanks for the rebuttal**
> >
> > Thanks for the response. Although the novelty seems to be a concern due to the simple combination of (Over)LORD and MoCo, to me such simplicity is a strength and finding the right combination that works well in practice is impactful, especially because there are so many different ways to combine things on such large datasets. But as noted by the authors, the motivation is not clear and more experiments could be done to help us understand why incorporating contrastive learning improves the results that much. Thus, I will keep my score as "accept".

---

### Official Review · Reviewer_Ex2Y · 2021-11-01

**Correctness:** 3
**Technical Novelty And Significance:** 1
**Empirical Novelty And Significance:** 3
**Recommendation:** 5
**Confidence:** 3

**Main Review:**

Strength:
1. The experiments successfully demonstrated that contrastive learning is somehow very useful in improving disentangled and discriminative representation learning. The improvement in terms of disentanglement learning in both synthetic and real datasets seems significant.
2. The ablation studies were helpful to convince that contrastive learning on encoders is critical in improving the disentanglement learning performance. Yet, it is still unclear how and why it improves the performance (see the comments below).

Weakness:
1. Despite the impressive results, the technical contribution of this work is still quite incremental. It basically follows exactly the same framework of (Over)LORD (latent optimization followed by the encoder training) but combines MoCo in the encoder training for disentangled factors. It is an interesting observation that this simple combination can lead to significant improvement in disentanglement learning, but the paper could be stretched in many ways instead of a naive combination (e.g., how latent optimization and contrastive learning can mutually improve each other).

2. The writing is somewhat unclear and not self-contained. For instance, (1) I assume that the paper follows the same problem setting of (Over)LORD where the class supervision is given. However, it is never mentioned clearly in the paper. It could be even more confusing since the discriminative embedding is indexed by the data instance, not clearly showing that instances in the same class share the same discriminative embedding. (2) Section 3.2 is not self-contained. The authors can describe why Eq.(1) can lead to disentanglement learning. (3) The authors did not mention if the generator parameters are trained in the first and/or second stages end-to-end. (4) It is unclear how the authors choose the positive pair $g^c_i$ in Eq.(4) and (5). Overall, I felt that the paper needs significant work to make descriptions clear and self-contained.

3. The objective of (Over)LORD and contrastive learning seem conflicting in some ways. (Over)LORD disentangles factors by constraining information within the latents (g) via Gaussian noise (Eq.(1)). On the other hand, contrastive learning encourages the same latent to be maximally informative for instance discrimination, which aids better generalization. A naive combination of two objectives seems to be invasive to each other, and it is unclear why it leads to improvement. It would be great to hear the authors’ thoughts on this.

4. It is not crystal clear the source of improvement in the encoders; the ablation study demonstrated that the improvement is mainly from the contrastive loss, but it is unclear if (1) the contrastive loss simply improves the generalization of the encoders in unseen images or (2) it also helps to discover disentangled factors in training data. My thought is that the latter is less likely since it shares nearly the same latent embeddings with (Over)LORD to provide supervision to the encoders (Eq.(3)). Evaluation of disentanglement learning in training/testing data could give us more insights. It would be great if the authors also share a more in-depth analysis of the source of improvement in the rebuttal.

5. In Table 1, comparisons to MoCO and SinCLR are unfair since they are self-supervised while the proposed method is based on supervised learning (i.e., trained with class labels). It should be clearly mentioned in the paper.

6. Qualitative analysis/comparisons could be very helpful but are missing in the paper. For instance, comparisons with baselines such as disentanglement performance, generation quality/diversity, etc. could be very useful.


**Summary Of The Paper:**

This paper proposed to improve the generative disentangled representation learning of LORD and OverLORD with contrastive learning. The idea is simple: when learning the amortized encoders in LORD and OverLORD, regularize them using contrastive learning to improve the generalization. Despite the simplicity, the authors demonstrated that this leads to a significant improvement in the disentanglement of representation.

**Summary Of The Review:**

Overall, I was impressed that simple contrastive learning can improve disentanglement learning. However, the paper did not go much deeper into why and how it improves the performance, and lack some technical novelty. The presentation should be improved in many ways too.

---

> ### Author Response · Authors · 2021-11-23
> **Response**
>
> We appreciate the insightful feedback, below we address the major concerns mentioned in the comments.
>
> Q1: Technical contribution.
>
> The main contribution of this work is to incorporate contrastive learning into latent optimization for both representation disentanglement and linear classification. We focus on demonstrating the effectiveness of contrastive learning in the disentanglement scenario, but not designing a working system. Therefore, we choose the LORD as a strong baseline and employ the most commonly used contrastive learning framework. On the other hand, we also reform the framework to leverage content-level and residual-level contrastive losses to capture content and residual representations across a large number of samples.
>
> Q2: Unclear points.
>
> (1) As we mentioned at the bottom of page 3, class supervision is given during latent optimization.
> (2) For Section 3.2, we follow the same setting in LORD except that we use two separate variables for learning content and residual embeddings.
> (3)  The generator parameters are trained in the first stage end-to-end. (4) The content-level contrastive loss in Eq.(4) is applied to discriminate content embeddings between the anchor and the negative samples. In contrast, the residual-level loss in Eq.(5) is designed to capture the invariant residual embeddings of the anchor.
>
> Q3: Reason for improvements.
>
> As illustrated in Figure 2, we maintain two separate momentum encoders for learning content and residual information during the encoder pre-training. The content-level contrastive loss is applied to discriminate content embeddings between the anchor and the negative samples, while the residual-level loss is designed to capture the invariant residual embeddings of the anchor. During the process of latent optimization, the content embeddings do not have noise added to them, but the residual embedding has noise added to it. In this way, we leverage the learned content embeddings and residual embeddings as the ''target'' in the encoder pre-training.

---

> > ### Comment · Reviewer_Ex2Y · 2021-11-29
> > **Response to rebuttal**
> >
> > I appreciate the authors' response. It partly addressed my concerns regarding some clarifications. However, I found that the response was not quite helpful to understand how the contrastive loss improves the performance; the authors did not address concerns regarding the conflicting objectives (Q3), generalization vs. disentanglement (Q4), and did not provide the qualitative analysis (Q6). I also found that some of the responses created more confusion; the author mentioned in the response that they did not add noise to the content embedding but to residual embeddings, but Eq.(1) shows that both are added with the noise. It is also not clear why the authors described the role of contrastive loss for content and residual embeddings differently although I could find meaningful differences in their equations (Eq.(4) and (5)). For these reasons, I maintain my score to be weak reject of this paper.

---

### Official Review · Reviewer_L4fH · 2021-11-06

**Correctness:** 3
**Technical Novelty And Significance:** 2
**Empirical Novelty And Significance:** 2
**Recommendation:** 5
**Confidence:** 2

**Main Review:**

The paper is well written and describes an interesting idea of augmenting MSE training with contrastive losses to improve the distilled latent representations of an encoder.

The experiments are quite extensive and include both synthetic and real-world datasets.

Additionally, I have a minor concern regarding the design of the model. The authors provide two types of learned continuous embeddings: content and residual, but from the text of the paper, I could not deduce the difference between these two variables, since they are treated equally throughout the optimization process and in the model itself. Therefore I would like the authors to clarify, what is the difference between these two variables.

The idea of using the two types of embeddings is not new and comes from another paper, which the authors site, but in this previous work, the two groups of variables are treated distinctly differently (namely, content embeddings do not have noise added to them, which allows learning representations correlated to the class labels).

Now to the main weakness.

The comparison with [4] is presented in Table 4 has mixed results and, to the best of my knowledge, no discussion within the paper. Does it seem like the proposed method is not directly superior to [4] when applied to real-world data? Why does the proposed method outperform [4] on synthetic benchmarks, but not on all real-world benchmarks?

Also, I consider the contribution of adding contrastive losses into the training of discriminative models to be quite minor. In the paper by Ojha et al. [1] the exact same idea was used to improve disentanglement using the InfoGAN base model. Yes, the authors of the paper under review use a different version of contrastive loss, but their exact realization is already a prior work and was described in a paper by He et al. [2].

It seems like the whole idea of the paper itself is the following: replacing disentangled learning part in paper [1] with [3, 4], and replacing the contrastive learning part with [2]. I would like the authors to describe their contributions more clearly and correct me if my impression is wrong.



[1] Utkarsh Ojha et al., "Elastic-InfoGAN: Unsupervised Disentangled Representation Learning in Class-Imbalanced Data"
[2] He et al., "Momentum Contrast for Unsupervised Visual Representation Learning"
[3] Gabbay et al., "DEMYSTIFYING INTER-CLASS DISENTANGLEMENT"
[4] Gabbay et al., "Scaling-up Disentanglement for Image Translation"

**Summary Of The Paper:**

The authors describe a method for learning disentangled representations (categorical and continuous). It is based on the generative latent optimization scheme for learning disentangled latent codes for a given dataset, followed by training an encoder network to predict these latent codes given an arbitrary image.

The main contribution of the authors is a proposition to include contrastive losses into the encoder training to prevent it from converging to a suboptimal state where it could predict similar embedding to dissimilar input images. They validate the resulting model using standard image generation and disentanglement benchmarks across multiple datasets.

**Summary Of The Review:**

While the idea describes in the paper is interesting, and the evaluation quite comprehensive, it seems like the essence of the authors' contribution itself is quite minor, and at the moment I am not convinced that it passes the acceptance threshold.

---

> ### Author Response · Authors · 2021-11-23
> **Response**
>
> We appreciate the insightful feedback, below we address the major concerns mentioned in the comments.
>
> Q1: Difference between content and residual variables.
>
> The main difference between these two variables is there are multiple content embeddings for capturing the information of each factor in an image, but only one residual embedding is applied to learn the residual information for this image. Thus, we do not have noise added to the content embeddings during latent optimization but have noises added to the residual embedding. In this way, we leverage the learned content embeddings and residual embeddings as the ''target'' in the encoder pre-training.
>
> Q2: Confusion about the added noise.
>
> Sorry for causing the confusion. The content embeddings should not have noise added to them, but the residual embeddings should have. We have updated them in the new manuscript.
>
> Q3: Comparison with OverLORD.
>
> Table 4 is presented to explore how each proposed loss affects the final performance of our method. For the comparison with OverLORD in Table 3, our ContraLORD achieves better performance on three real benchmarks in terms of 5 out of 7 evaluation metrics. We have the discussion at the ''Evaluation of disentangled embeddings'' part in the manuscript.
>
> Q4: Main contribution.
>
> The main contribution of this work is to incorporate contrastive learning into latent optimization for both representation disentanglement and linear classification. We focus on demonstrating the effectiveness of contrastive learning in the disentanglement scenario, but not designing a working system. Therefore, we choose the LORD as a strong baseline and employ the most commonly used contrastive learning framework. On the other hand, we also reform the framework to leverage content-level and residual-level contrastive losses to capture content and residual representations across a large number of samples.

---

> > ### Comment · Reviewer_L4fH · 2021-12-05
> > **Final review**
> >
> > I thank the authors for the comments, however, still remain not convinced on the lack of flaws I have mentioned previously in the review.
> >
> > The authors claim to have achieved similar or better scores compared to the OverLORD method. However, if we calculate the difference between the metrics that ContraLORD and OverLORD achieve, we arrive at a 3-4% difference, for the first two datasets, and <1% on the last dataset. Based on that, I find the discussion provided in the paper lacking: "We can observe that our ContraLORD achieves the best performance in terms of 5 out of 7 evaluation metrics. For the other two metrics, we still achieve comparable results when compared
> > to OverLORD". If the gap of 3-4% in terms of performance was found to be significant when ContraLORD outperforms OverLORD, then it should be considered significant in the opposite scenario, and the discussion should be provided.
> >
> > Could the difference in the values of these metrics be attributed to the different choice of losses or re-balancing on weights? Is the training schedule for these two methods similar, or one was simply trained for more iterations? Unfortunately, since the paper lacks this discussion, these questions cannot be answered using the data and insights provided in the manuscript. And these questions are important when one wants to choose which method to apply in practice.
> >
> > Regarding the novelty, the authors claim to have explored the "effectiveness of contrastive learning in the disentanglement scenario, but not designing a working system". However, the authors only incorporate contrastive learning into a single method (ContraLORD). Therefore, the paper provides no insights into the general effectiveness of contrastive learning for the problem of disentangled representations learning. To show it, they would require to incorporate the proposed training method into the alternative systems and show its effectiveness for all baselines. Therefore, I respectfully can only consider the contribution of this paper to be a new and very specific system, which, as I have stated in the main review, is, in my opinion, quite lacking.
> >
> > Based on all of that, I maintain my recommendation (marginally below the acceptance threshold).

---

### Official Review · Reviewer_kr5R · 2021-11-08

**Correctness:** 3
**Technical Novelty And Significance:** 2
**Empirical Novelty And Significance:** 3
**Recommendation:** 5
**Confidence:** 4

**Main Review:**

Strengths:
1. The experiment results are convincing. The proposed method outperforms strong baselines and obtains state-of-the-art performance. The ablation study is comprehensive and clearly shows the empirical improvement brought by the proposed techniques.

Weaknesses:
1. The proposed method is a simple combination of LORD followed by contrastive representation learning. The technical novelty is limited.
2. The motivation of using contrastive learning to learn disentangled representation is unclear. The goal of contrastive learning is to learn embeddings that are invariant to certain data augmentations, while the goal of disentanglement is to learn embeddings that are invariant to label-unrelated generative factors. The data augmentations are not necessarily label-preserving. Also, there exist label-unrelated variations not captured by data augmentations. It is not clear to me why contrastive learning helps disentanglement.
3. The presentation is not very clear and I find it hard to understand some parts of the algorithm. Section 3.3 does not mention any usage of data augmentation during encoder pre-training, while section 4 states the same data augmentation as in MoCo is used. Authors should clarify what data augmentations are applied to which inputs during the encoder pre-training. Besides, it seems the content embedding and the residual embedding are trained with the same objective. It is not clear to me how they learn different information. The purpose of having three embeddings instead of two (as in LORD) is also not well explained.

Minor comments that do not affect the rating:
1. Section 3.1 "discriminative embeddings d" should be "d_i"
2. "The InfoNCE loss, that is, the normalized temperature-scaled cross-entropy loss, are " -> "is"

**Summary Of The Paper:**

The paper proposes a new two-step approach for learning disentangled representation. In the first step, a generator is learned to map latent embeddings to reconstruct images. Similar to LORD, the embedding corresponding to each image is jointly optimized with the generator weights. The second step uses contrastive learning together with regression loss to learn an encoder that maps images to the latent space, which is the main contribution of the paper. Results show that the proposed approach achieves state-of-the-art performance on several datasets for disentangled representation learning.

**Summary Of The Review:**

Despite strong empirical performance, I am incline towards rejection due to the limited novelty and unclear motivation. The presentation can be improved as well.

---

> ### Author Response · Authors · 2021-11-23
> **Response**
>
> We appreciate the insightful feedback, below we address the major concerns mentioned in the comments.
>
> Q1: Novelty.
>
> The main contribution of this work is to incorporate contrastive learning into latent optimization for both representation disentanglement and linear classification. We focus on demonstrating the effectiveness of contrastive learning in the disentanglement scenario, but not designing a working system. Therefore, we choose the LORD as a strong baseline and employ the most commonly used contrastive learning framework. On the other hand, we also reform the framework to leverage content-level and residual-level contrastive losses to capture content and residual representations across a large number of samples.
>
> Q2: Motivation of contrastive learning.
>
> As shown in Figure 2, we maintain two separate momentum encoders for learning content and residual information during the encoder pre-training. The content-level contrastive loss is applied to discriminate content embeddings between the anchor and the negative samples. In contrast, the residual-level loss is designed to capture the invariant residual embeddings of the anchor.
>
>
> Q3: Confusion about data augmentations.
>
> Sorry for causing the confusion. For the comparison with previous self-supervised methods in terms of linear classification on CIFAR-10, CIFAR-100, and ImageNet-100, we apply the same data augmentation as in MoCo is used. This data augmentation includes RandomResizedCrop, RandomGrayscale, RandomHorizontalFlip.
>
> Q4: Differences between content and residual embeddings.
>
> As we explained in Q2, we maintain two momentum queues for the query sample to capture the anchor's invariant content-level and residual-level embeddings. Thus, during latent optimization, the content embeddings do not have noise added to them, but the residual embedding has noise added to it. In this way, we leverage the learned content embeddings and residual embeddings as the ''target'' in the encoder pre-training.

---

### Decision · Program_Chairs · 2022-01-20

**Decision:**

Reject

**Comment:**

The paper presents modifying latent optimization for representation disentanglement using contrastive learning, resulting in improved performance on disentanglement benchmarks. Despite the empirical success, the proposed algorithm has many moving parts and loss functions. Most reviewers agree that given the incremental and complex nature of the proposed technique, the empirical results are not sufficient for acceptance at ICLR, especially since the results do not present additional insights into the inner workings of the method. I encourage the authors to try to simplify the technique, or provide a convincing evidence that such complexity is necessary.

PS:
I didn't find much discussion of how the hyper-parameters are chosen (temperature, lambda terms, etc.).
A discussion of recent self-supervised disentanglement methods (e.g., https://arxiv.org/abs/2102.08850 and https://arxiv.org/abs/2007.00810) can be helpful.